# Machine Learning Based Power Quality Disturbance Classification Under Varying Fundamental Frequency Conditions for Islanding of a Microgrid

## Abstract

The integration of renewable energy sources into modern power systems is increasing in a rapid manner. With their integration into the grid and many other power electronics devices, the rise in Power Quality Disturbances (PQDs) is posing as a major issue in improving the performance of smart grids and microgrid systems. Due to the intermittency of renewable energy sources, the fundamental frequency of power systems fluctuate, stemming different PQDs. In this work, we present a comparative analysis of four different machine learning classification algorithms, namely Support Vector Machine (SVM), k-Nearest Neighbours (KNN), Random Forest (RF), and Naive Bayes (NB), in classifying different types of single-stage and multiple-stage PQDs. We generated a dataset consisting of 29 types of PQDs, while taking into consideration the variation of fundamental frequency. We note that variation of fundamental frequency can significantly affect the classification accuracy of the machine learning models. We introduced different levels of noise (20dB to 40dB) to the PQD data to mimic real-life scenarios. We used the Linear Discriminant Analysis (LDA), which is a supervised learning technique, for dimensionality reduction and feature extraction. Our results show that the classification accuracy decreases as the noise level is increased gradually, and that a decrease in the classification accuracy is possible when the variation of the fundamental frequency is taken into account.

## 1 Introduction

The growing use of renewable energy and its connection to the grid has led to a surge in power semiconductor devices. Such integration of renewable energy sources, driven by rising energy demands, increases power quality disturbances (PQDs) that cause significant issues in distribution systems. In the present scenario of power systems, microgrids are a very popular and emerging idea. Microgrids are especially prone to PQDs due to high integration of renewable energy sources and their intermittencies (Hatziargyriou et al., 2007). Additionally, the boom in power electronics technology has brought a surge in non-linear loads, negatively impacting power quality. These loads, while beneficial, contribute to power quality (PQ) problems like capacitor bank failures, increased distribution and equipment losses, and motor vibrations. Ideally, the power system should deliver pure sine waves, but power electronic converters distort this waveform. To address these issues, standards, such as the IEEE 519, have been established to regulate the harmonic levels within the power system. Nevertheless, the need for effective solutions to mitigate these power quality disturbances.

Numerous incidents of PQDs can cause inconvenience and economic losses to users. For the mitigation of PQDs, an uninterrupted monitoring of power signals is needed, which is nontrivial. In this context, it is essential to efficiently, accurately, and quickly classify PQDs. To this end, light-weight classification algorithms can be employed for the real-time monitoring system of the power grid. PQDs can consist of stationary signals (flickers), non-stationary signals (oscillatory transients and notches), and a combination of both. As such, PQD identification incorporates feature extraction of PQDs, and classification using the extracted features. The selection of relevant features is instrumental for accurate classification. Therefore,

several feature extraction tools are employed, including, but not limited to Fast Fourier Transform (FFT) (Lopez-Ramirez et al., 2018), Stockwell Transform (ST) (Kumar et al., 2022), Wavelet Transform (WT) (Yılmaz et al., 2022), Gabor Transform (GT) (Liu et al., 2018), Hilbert Transform (HT) (Granados-Lieberman et al., 2013), Empirical Mode Decomposition (EMD) (Camarena-Martinez et al., 2015), Short Time Fourier Transform (STFT) (Singh & Singh, 2017), Singular Value Decomposition (SVD) (Lopez-Ramirez et al., 2018), and Kalman Filter (KF) (He et al., 2022).

In this work, we propose a new method for classifying both single and combined PQDs of a microgrid system using a simultaneous dimensionality reduction and feature extraction approach. Our approach aims to transform complex data with many features into a simpler form with fewer, but more relevant, features. More specifically, our proposed method utilizes Linear Discriminant Analysis (LDA), which is a supervised learning technique that focuses on maximizing the separation between different classes, while minimizing variations within each class (Xanthopoulos et al., 2012). LDA also allows for the efficient reduction of data dimensionality. We generate a synthetic dataset consisting of different PQDs under varying fundamental frequency conditions (48 Hz to 52 Hz with a 0.5 Hz increment), so that our proposed approach is more robust. We note that the variation of the fundamental frequency can occur in practical systems due to the islanding of the microgrid from the main grid. We evaluated the performance of LDA-based feature extraction using four machine learning (ML) classifier models: k-Nearest Neighbor (KNN), Naive Bayes (NB), Support Vector Machine (SVM), and Random Forest (RF). Our results show that LDA-based feature extraction and dimensionality reduction achieves excellent classification accuracy for 29 different PQD types, even with varying noise levels (20 dB to 40 dB).

## 2 Literature Review

There are a number of existing works that used various ML models for extracting features and classifying PQDs. Dhote et al. (2015) showed different ways of recording PQD data from MATLAB-SIMULINK environment, whereas Ray et al. (2019) demonstrated a microgrid model, and extraction of 6 PQDs. They also classified the data using Independent Component Analysis (ICA) and SVM. Vinayagam et al. (2021) investigated PQD detection and classification from PV integrated microgrids. Chakravorti et al. (2018) proposed real-time detection and classification of 10 types of PQDs, whereas Suganthi et al. (2021) demonstrated the detection and classification of 8 types of PQDs from microgrid networks. Recently, Singh et al. (2023) demonstrated the classification of 29 types of PQDs using LDA on a synthetic dataset generated according to the works of Igual et al. (2018). We note that Zhang et al. (2011), Gong & Ruan (2020), Xue et al. (2020), and Wang & Chen (2019) also generated data from mathematical models of microgrid.

Several classification techniques have been applied by the researchers for the classification of PQDs in existing literature. Among classical ML methods, Kernel ridge regression was utilized by Chakravorti et al. (2019) for classification purposes, whereas Rule-based decision trees were implemented in Igual et al. (2019) and Zhang et al. (2011). Singh et al. (2023) presented results of multiple algorithms, such as SVM, RF, NB, and KNN, for the classification. A Convolutional Neural Network (CNN) was used for the classification in Gong & Ruan (2020), Xue et al. (2020), and Wang & Chen (2019). DWT, Bayesian net, Decision tree and MLP, were used in Vinayagam et al. (2021), and UWT and SGBT were used in Yılmaz et al. (2022).

Most of the existing works used the grid fundamental frequency to test the classification systems, ignoring frequency deviations. In practice, variation of the fundamental frequency can occur due to the islanding of the microgrid from the main grid. Igual et al. (2019) tested a rule-based decision tree for classification under different fundamental frequencies and showed a comparison of the accuracies for the variation of frequencies *in different datasets*. However, in weak islanded microgrids with low inertia, there can be significant changes in fundamental frequency (Igual et al., 2019), which most of the existing studies ignored. To address this research gap, we evaluate the performance of ML classifiers for PQD classification in microgrids operating in islanded mode. More specifically, we investigate the robustness of ML classification algorithms if they are trained on a dataset containing different variations of fundamental frequencies, which is common for most grid-connected systems or strong islanded microgrids.

## 3 Background and Problem Formulation

**Notation.** The vectors, matrices, and scalers are denoted with bold lowercase letters (e.g. $\mathbf{x}$), bold uppercase letters (e.g., $\mathbf{X}$), and unbolded letters (e.g., $N$), respectively. The indices are denoted with unbolded lowercase letters. The $n$-th column of the matrix $\mathbf{X}$ is denoted as $\mathbf{x}_n$. We sometimes denote the set $\{1, 2, \ldots, N\}$ with $[N]$.

### 3.1 Dimensionality Reduction

A commonly-used approach to reduce the dimensionality of the samples of a dataset is Linear Discriminant Analysis (LDA). LDA is a supervised learning technique that aims to project the samples onto a lower-dimensional subspace, where the classes are well separated (Xanthopoulos et al., 2012).

Consider a dataset $(\mathbf{X}, \mathbf{y})$, where $\mathbf{X} = [\mathbf{x}_1, \mathbf{x}_2, ..., \mathbf{x}_N] \in \mathbb{R}^{d \times N}$ contains $N$ samples or data points, and corresponding class label vector $\mathbf{y} = [y_1, y_2, \ldots, y_N]^\top \in \mathbb{R}^N$. Here, each sample $\mathbf{x}_n$ is a $d$-dimensional feature vector and $y_n$ indicates the class label for data point $\mathbf{x}_n$. Now, the Within-Class Scatter Matrix $\mathbf{S}_w$ and the Between-Class Scatter Matrix $\mathbf{S}_b$ are calculated as follows:

$$\mathbf{S}_w = \sum_{c=1}^{C} \mathbf{S}_w^c, \tag{1}$$

where $C$ is the number of classes, and $\mathbf{S}_w^c$ is the scatter matrix for class $c \in [C]$, defined as:

$$\mathbf{S}_w^c = \sum_{i \in \text{Class } c} (\mathbf{x}_i - \boldsymbol{\mu}_c)(\mathbf{x}_i - \boldsymbol{\mu}_c)^\top, \tag{2}$$

where $\boldsymbol{\mu}_c \in \mathbb{R}^d$ represents the mean vector of class $c$. The $\mathbf{S}_w$ matrix captures the variability within each class (Xanthopoulos et al., 2012). Next, the Between-Class Scatter Matrix $\mathbf{S}_b$ captures the separation between different classes. It is defined as:

$$\mathbf{S}_b = \sum_{c=1}^{C} N_c (\boldsymbol{\mu}_c - \boldsymbol{\mu})(\boldsymbol{\mu}_c - \boldsymbol{\mu})^\top, \tag{3}$$

where $N_c$ is the number of data points in class $c$, and $\boldsymbol{\mu} \in \mathbb{R}^d$ is the overall mean vector of the entire dataset. Now, a set of $d_p$ projection vectors (where $d_p \leq d$ is the desired dimensionality) is sought that maximizes the ratio of the between-class scatter to the within-class scatter. To that end, we form a transformation matrix $\mathbf{W} = [\mathbf{w}_1, \mathbf{w}_2, ..., \mathbf{w}_{d_p}] \in \mathbb{R}^{d \times d_p}$, such that the following holds

$$\mathbf{S}_b \mathbf{w}_i = \lambda_i \mathbf{S}_w \mathbf{w}_i \text{ for } i \in [d_p], \tag{4}$$

where $\{\lambda_i\}$ are non-negative scalers. Typically, the vectors $\{\mathbf{w}_i\}$ are chosen in decreasing order of their corresponding $\{\lambda_i\}$ values. This results in the projection vectors that capture the most significant class separation information. Projecting the original data points onto $\mathbf{W}$ subspace yields a lower-dimensional representation

$$\mathbf{Z} = \mathbf{W}^\top \mathbf{X}, \tag{5}$$

where $\mathbf{Z} \in \mathbb{R}^{d_p \times N}$ is the reduced-dimensional data matrix. Finally, a classifier can be trained on this transformed data.

### 3.2 Classification Methods

#### 3.2.1 Support Vector Machine

Support Vector Machine (SVM) is one of the most extensively used classical machine learning techniques that is capable of solving classification problems. Although mathematically complex and computationally expensive, SVM can attain excellent accuracy in classification for the PQDs (Gandhi, 2018). SVM works by finding the best decision boundary or hyperplane to categorize $d$-dimensional samples into classes.

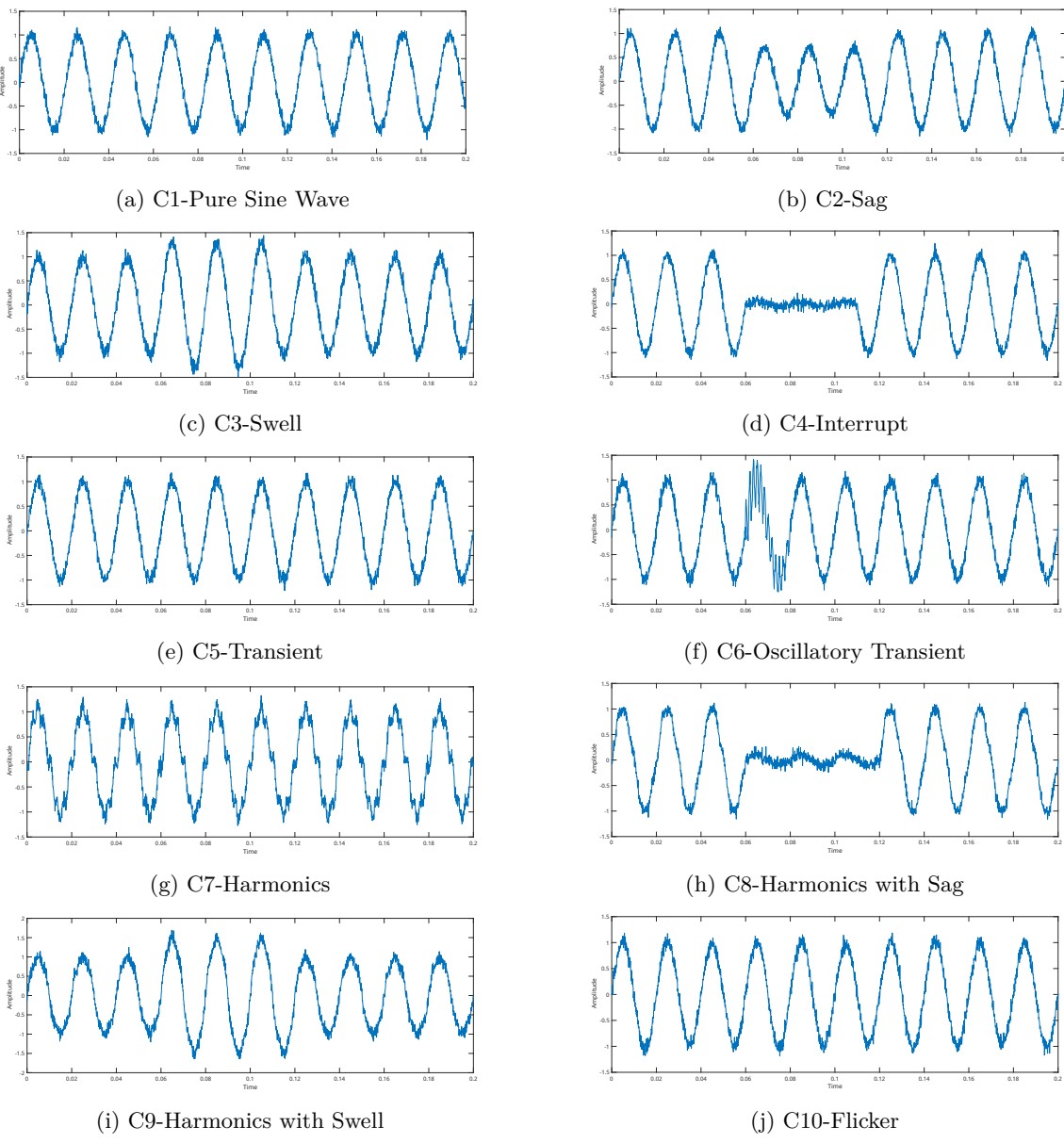

Figure 1: Sample of different PQDs with 20dB noise

In SVM, if the labels are considered as $y_n \in \{-1, 1\}$, the maximum margin hyperplane that separates the two groups of data points is $\mathbf{w}^\top \mathbf{x} - \mathbf{b} = \mathbf{0}$. Here, $\mathbf{w} \in \mathbb{R}^d$ is the normal vector to the hyperplane, and $b$ is a scalar. Along $\mathbf{w}$, the offset of the hyperplane from the origin is determined by the parameter $\frac{b}{\|\mathbf{w}\|}$. Here, $\|\mathbf{w}\|$ has to be minimized in order to maximize the distance between the planes. Hence, the optimization for SVM can be written as,

$$\underset{\mathbf{w}, b}{\text{minimize}} \ \|\mathbf{w}\|_2^2 \quad \text{subject to} \quad y_n(\mathbf{w}^\top \mathbf{x}_n - b) \geq 1, \forall n \in [N]. \tag{6}$$

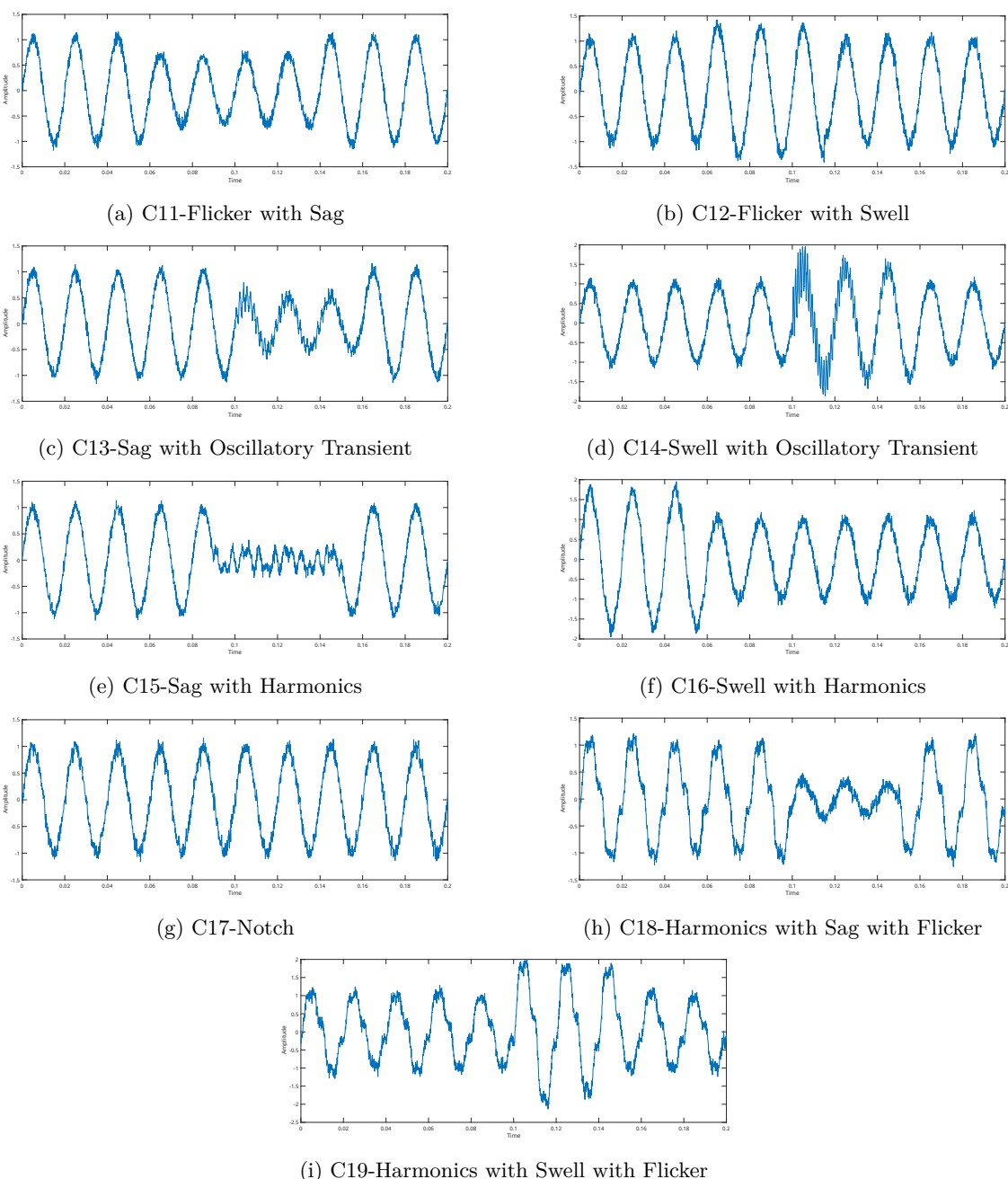

(a) C11-Flicker with Sag

(b) C12-Flicker with Swell

(c) C13-Sag with Oscillatory Transient

(d) C14-Swell with Oscillatory Transient

(e) C15-Sag with Harmonics

(f) C16-Swell with Harmonics

(g) C17-Notch

(h) C18-Harmonics with Sag with Flicker

(i) C19-Harmonics with Swell with Flicker

Figure 2: Sample of different PQDs with 20dB noise

### 3.2.2 K Nearest Neighbors

K Nearest Neighbors (KNN) is an intuitive non-parametric machine learning algorithm for classification and regression tasks. It classifies a new data point based on the majority vote of its $K$ nearest neighbors in the training data (Guo et al., 2003). Although no explicit training is necessary, the algorithmic process of KNN can be described as:

To formulate the KNN methodology, considering $\mathbf{x} \in \mathbb{R}^d$ be a data point, and $\mathbf{X} = [\mathbf{x}_1, \mathbf{x}_2, \ldots, \mathbf{x}_N] \in \mathbb{R}^{d \times N}$ be a set of training data points. Theoretically, the $K$-nearest neighbors of $\mathbf{x}_n$ are the $K$ points in $\mathbf{X}$ that have the smallest Euclidean distance to $\mathbf{x}_n$. The Euclidean distance $s$ between two points $\mathbf{x}_i$ and $\mathbf{x}_j$ is

$s(\mathbf{x}_i, \mathbf{x}_j) = \|\mathbf{x}_i - \mathbf{x}_j\|_2$. Now, if there is a set of labeled training data $\{(\mathbf{x}_1, y_1), (\mathbf{x}_2, y_2), \ldots, (\mathbf{x}_N, y_N)\}$, where $y_n$ is the label of $\mathbf{x}_n$, the KNN algorithm assigns the label to an *unseen* $\mathbf{x}$ based on the majority vote of its K-nearest neighbors. That is, if $\mathcal{N}_K(\mathbf{x})$ is the set of the K-nearest neighbors of $\mathbf{x}$, the predicted label $\hat{y}$ for $\mathbf{x}$ is:

$$\hat{y} = \arg\max_y \sum_{\mathbf{x}_i \in \mathcal{N}_K(\mathbf{x})} \mathbb{I}(y_i = y),$$

where $\mathbb{I}(\cdot)$ is the indicator function.

### 3.2.3 Random Forests

Random Forest (RF) is an ensemble learning technique. They are very powerful in tasks, such as regression and classification. By combining multiple decision trees, RF can achieve improved performance and robustness compared to the single decision trees (Liaw et al., 2002). A standard decision tree is a hierarchical structure where each node is split using the best split among all the available variables. But in a RF, each node is split using the best split among a subset of randomly chosen predictors. Each tree in the forest casts a vote for a particular class, and the final prediction is the majority vote among all the trees.

### 3.2.4 Naive Bayes

Naive Bayes (NB) is a part of the probabilistic generative classifiers. The part *Naive* in its name comes from its assumption of the independent occurrence of features, and the *Bayes* part comes from the fact that the algorithm is based on Bayesian theories. The Bayes' theorem states:

$$P(C_i|\mathbf{x}) = \frac{P(\mathbf{x}|C_i)P(C_i)}{P(\mathbf{x})} \tag{7}$$

where, $P(C_i|\mathbf{x})$ is the posterior probability of class $C_i$ given the data point $\mathbf{x}$, which is to be calculated for classification. For a new data point $\mathbf{x}$, Naive Bayes calculates the posterior probability for each class $C_i$ using Bayes' theorem. The class with the highest posterior probability is predicted as the class label for the new data point (Rish et al., 2001). We employed the Gaussian Naive Bayes for classification purposes.

## 4 Dataset Generation

**Specifics of the Microgrid.** The mathematical model described in Igual et al. (2018) formulated twenty-nine different types of disturbances, including 8 unique disturbances and 21 combined disturbances. All of these disturbances are defined with a set of parameters. We generate our dataset using this model, which is implemented in MATLAB. The disturbances correspond to specific variables, that can be varied according to the IEEE guidelines (IEEE, 2009) to generate variation within the data.

**PQD Data.** To generate the dataset for classification, a total of 29 types of PQDs were generated. The PQDs were generated in MATLAB as per the guidelines described in IEEE-1159 standards, which is widely used for simulating PQD data that closely portrays the real-world scenario. Note that collecting real-world PQD data is nontrivial, since it requires monitoring and assessment of power grid events for a long time as the instance when a PQD might occur in a system is random. To create the dataset, the variation of fundamental frequency from 48Hz to 52Hz with an increment of 0.5Hz was considered, as microgrid systems experience frequency fluctuation during the islanding process. The different PQDs were generated randomly using the mathematical model described in Igual et al. (2018) with varying parameters. The core parameters that were maintained throughout the generation of all the PQDs are given in Table 1.

To make the data replicate real-world data more closely, noise of different levels (namely, SNR 20dB, 25dB, 30dB, 35dB, and 40dB), were added to the samples. A total of 6 different datasets were generated to compare the accuracy of classification based on different noise levels. The different types of PQDs with 20dB noise levels are shown in Figures 1, 2, and 3.

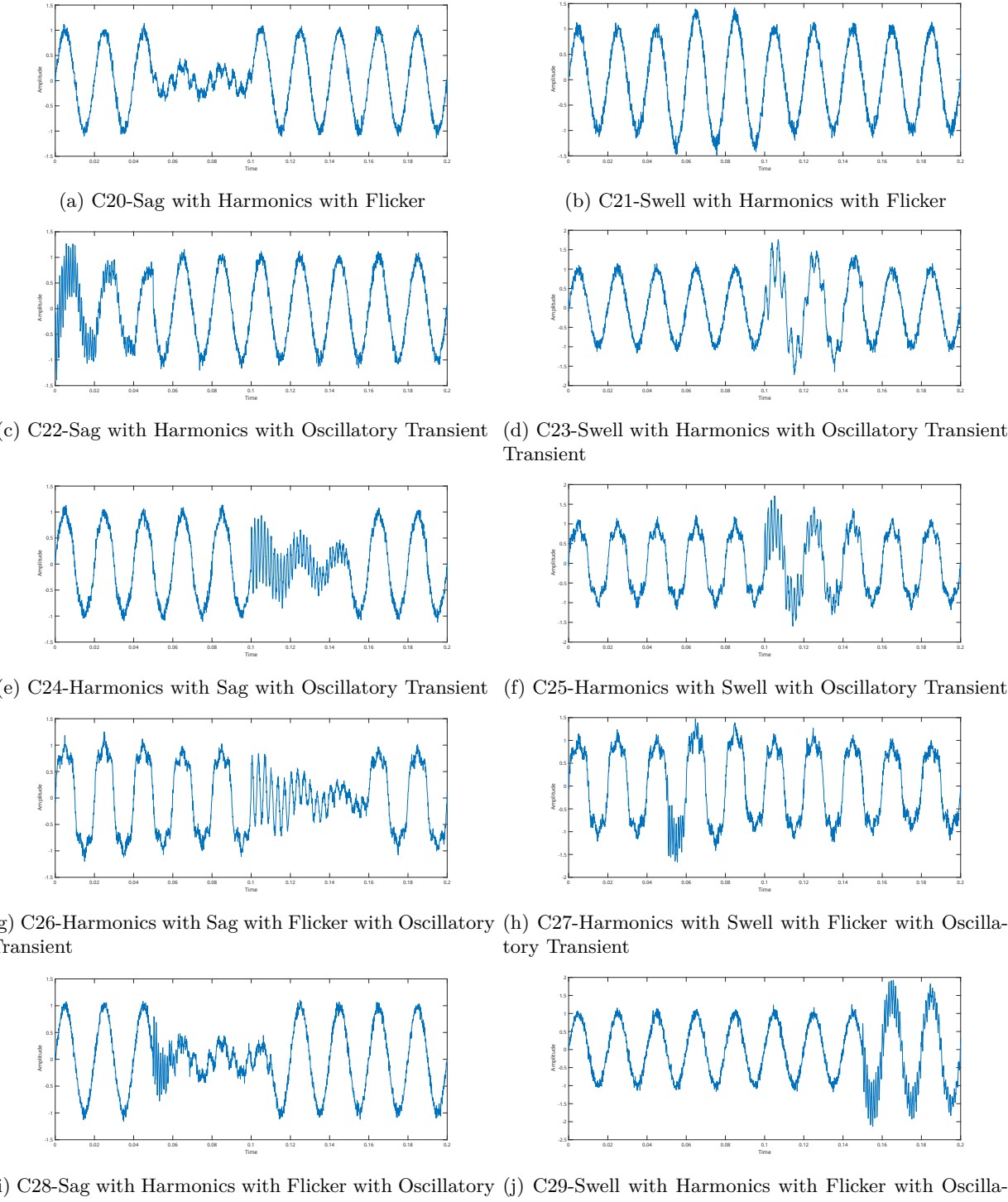

(a) C20-Sag with Harmonics with Flicker  (b) C21-Swell with Harmonics with Flicker

(c) C22-Sag with Harmonics with Oscillatory Transient  (d) C23-Swell with Harmonics with Oscillatory Transient Transient

(e) C24-Harmonics with Sag with Oscillatory Transient  (f) C25-Harmonics with Swell with Oscillatory Transient

(g) C26-Harmonics with Sag with Flicker with Oscillatory Transient  (h) C27-Harmonics with Swell with Flicker with Oscillatory Transient

(i) C28-Sag with Harmonics with Flicker with Oscillatory Transient  (j) C29-Swell with Harmonics with Flicker with Oscillatory Transient

Figure 3: Sample of different PQDs with 20dB noise

## 5 Experimental Results

As mentioned before, 29 types of PQDs were generated using the mathematical model (Igual et al., 2018) with random variation of the given parameters. Introducing the frequency variation, 900 samples were

| Parameter | Value |
|---|---|
| Fundamental Frequency Variations | 48Hz to 52Hz with an increment of 0.5Hz |
| Samples per Frequency per Disturbance | 100 |
| Total Samples per Disturbance | 900 (100 samples/frequency $\times$ 9 frequencies) |
| Amplitude | 1.0 pu |
| Sampling Frequency | 16 kHz |
| Cycles per Sample Signal | 10 |

Table 1: Parameters used for generating PQD data.

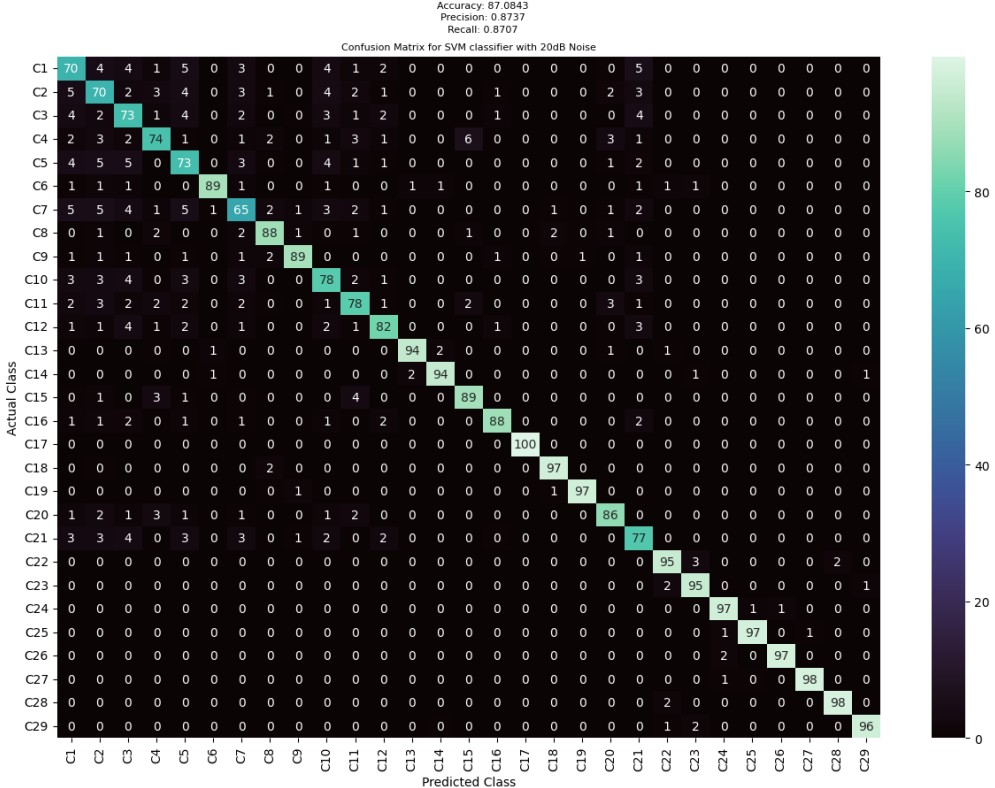

Figure 4: Confusion matrix for SVM classifier with 20dB noise

generated for each of the 29 classes. Ten cycles for each class were considered, and the sampling rate was set to 16kHz. The fundamental frequency was varied from 48Hz to 52Hz, and white Gaussian noise was added with SNR ranging from 20dB to 40dB.

**Dimension Reduction using LDA.** LDA is used dimensionality reduction, and simultaneously extract features from the raw samples. As mentioned before, LDA projects the data samples onto lower dimensional subspace that enhances classification performance. After using the LDA, the sample dimension $d$ was reduced from 3200 to $d_p = 28$. These reduced-dimensional samples were used for the training the classifier. These samples are split into train-test split (with 80:20 ratio).

**Classification Performance.** In Table 2, the classification performance for different algorithms under different levels of noise are shown. It can be observed that, as the noise level is increased, the classification accuracy starts to decrease. Additionally, when the signal is without any noise, RF algorithms gives the best

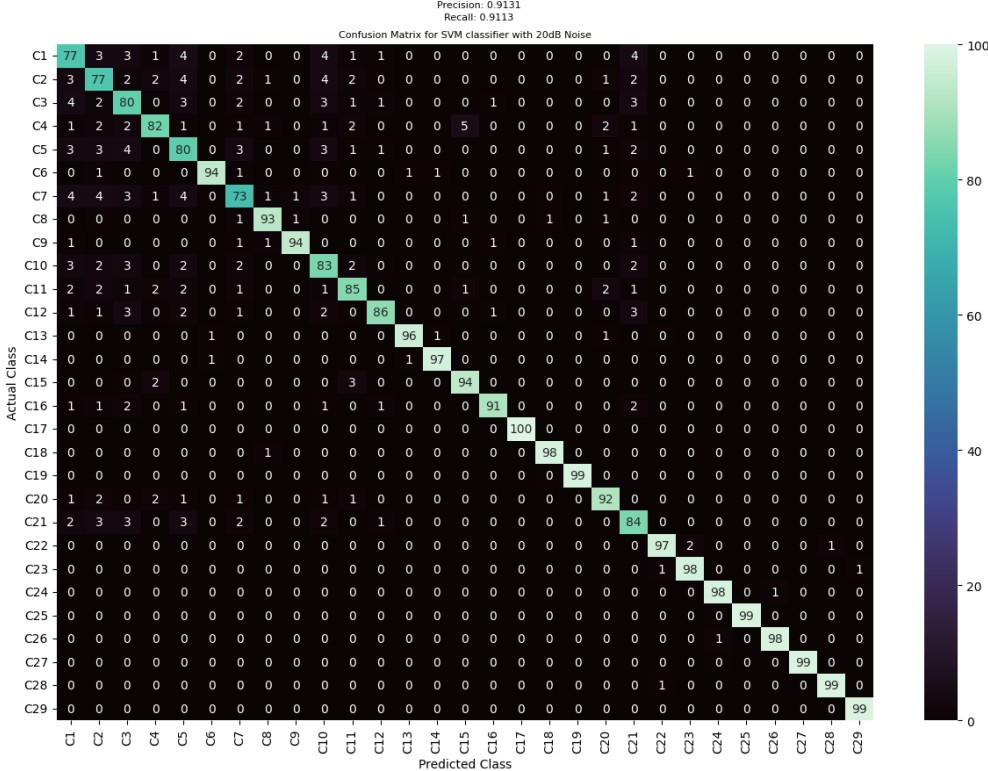

Figure 5: Confusion matrix for kNN classifier with 20dB noise

| Classification algorithm | Without Noise | 40dB | 35dB | 30dB | 25dB | 20dB |
|---|---|---|---|---|---|---|
| SVM | 0.9483 | 0.9079 | 0.8985 | 0.8919 | 0.8826 | 0.8708 |
| KNN | 0.9841 | 0.9898 | 0.8587 | 0.8408 | 0.8271 | 0.8003 |
| RF | 0.9896 | 0.8939 | 0.8843 | 0.8754 | 0.8629 | 0.8453 |
| NB | 0.9145 | 0.8877 | 0.8800 | 0.8772 | 0.8711 | 0.8539 |

Table 2: Accuracy for different classification algorithms.

accuracy. But as noise is introduced in the dataset, SVM classifier shows a consistent result in classification. We show the confusion matrices for the four classification algorithms in Figures 4-7.

Note that, the authors in Singh et al. (2023) used LDA to classify 29 types of PQDs, and the findings show somewhat counter-intuitive results – when the noise level was increased, the accuracy of the classifier also increased. But in this current study, the finding shows an intuitive result where as the noise level is increased, the accuracy decreases gracefully. To the best of out knowledge, no other existing work has investigated the variation of fundamental frequency in a single dataset, and hence, our results cannot be compared with any other existing literature.

## 6 Conclusion

In this work, we explored classification of power quality disturbances for microgrids, where the fluctuation in the fundamental frequency during the islanding process has been taken into consideration. The significance of the work is underlined by considering the fact that mitigating system failures and malfunctioning of electrical elements connected to a microgrid is of tremendous importance for power system stability and performance.

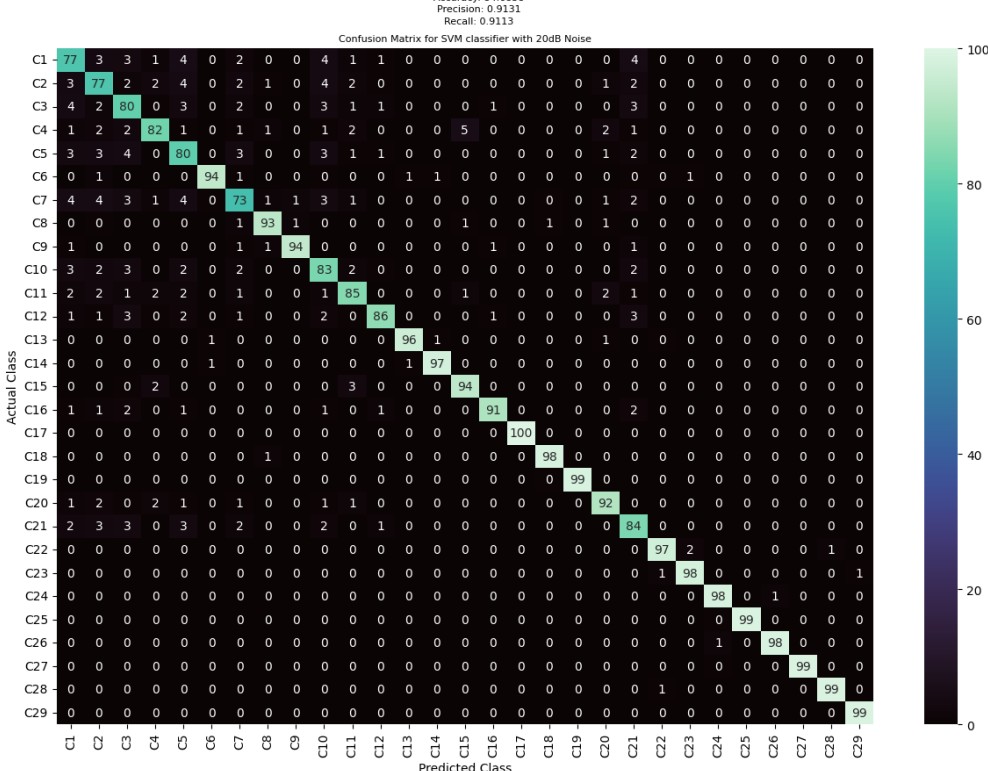

Figure 6: Confusion matrix for Random Forest classifier with 20dB noise

Our results clearly show a significant decrease in the classification accuracy when this variation is taken into account. There is sufficient scope to attempt to increase the accuracy score during this situation and explore other algorithms in this case. The supervised dimension reduction and feature extraction approach that was employed in this research was linear discriminant analysis, which paves a faster and simpler way in the classification procedure. There were 29 types of PQDs that were taken into consideration for the classification process, and 4 types of machine learning algorithms were used for this process. Different levels of noise were added to the sample signals to better replicate the synthetic data to real-world signals. The results show that, the SVM classifier had the best performance of the four in the classification in the presence of noise in the signals. To the best of our knowledge, our work is the first foray into investigating the variation of fundamental frequency in a single dataset of PQD. We believe that our work will pave the way of more efficient and accurate classification of real-world PQD.

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

Figure 7: Confusion matrix for Naive-Bayes classifier with 20dB noise

based microgrid. *Renewable energy focus*, 28:78–99, 3 2019. doi: 10.1016/j.ref.2018.12.002. URL `https://linkinghub.elsevier.com/retrieve/pii/S1755008418303156`.

Paramanand V. Dhote, B. T. Deshmukh, and B. E. Kushare. Generation of power quality disturbances using matlab-simulink. In *2015 International Conference on Computation of Power, Energy, Information and Communication (ICCPEIC)*, pp. 0301–0305, 2015. doi: 10.1109/ICCPEIC.2015.7259479.

Rohith Gandhi. Support vector machine—introduction to machine learning algorithms. *Towards Data Science*, 7(06), 2018.

Renxi Gong and Taoyu Ruan. A new convolutional network structure for power quality disturbance identification and classification in micro-grids. *IEEE Access*, 8:88801–88814, 2020. doi: 10.1109/ACCESS.2020.2993202.

David Granados-Lieberman, Martin Valtierra-Rodriguez, Luis A Morales-Hernandez, Rene J Romero-Troncoso, and Roque A Osornio-Rios. A hilbert transform-based smart sensor for detection, classification, and quantification of power quality disturbances. *Sensors*, 13(5):5507–5527, 2013.

Gongde Guo, Hui Wang, David Bell, Yaxin Bi, and Kieran Greer. Knn model-based approach in classification. In *On The Move to Meaningful Internet Systems 2003: CoopIS, DOA, and ODBASE: OTM Confederated International Conferences, CoopIS, DOA, and ODBASE 2003, Catania, Sicily, Italy, November 3-7, 2003. Proceedings*, pp. 986–996. Springer, 2003.

Nikos Hatziargyriou, Hiroshi Asano, Reza Iravani, and Chris Marnay. Microgrids. *IEEE power and energy magazine*, 5(4):78–94, 2007.

Hucheng He, Linke Wang, Chenghai Wang, Jingzhe Xu, and Kaikai Li. An identification method of power quality disturbances using kalman filter and extreme learning machine. In *2022 IEEE 5th Advanced Infor-*

*mation Management, Communicates, Electronic and Automation Control Conference (IMCEC)*, volume 5, pp. 1701–1705. IEEE, 2022.

IEEE. Ieee recommended practice for monitoring electric power quality, 2009.

Raul Igual, Carlos Medrano, and Franz Schubert. Evaluation of automatic power quality classification in microgrids operating in islanded mode. In *2019 IEEE Milan PowerTech*, pp. 1–6, 2019. doi: 10.1109/ PTC.2019.8810911.

Raúl Igual, Carlos Medrano, Francisco Javier Arcega, and Gabriela Mantescu. Integral mathematical model of power quality disturbances. pp. 1–6, 2018. doi: 10.1109/ICHQP.2018.8378902.

Rajat Kumar, Abhinav Saxena, Raj Kumar, Sanjay Marwaha, Jay Singh, and Gyanendra Kumar Singh. A comprehensive overview on modified versions of stockwell transform for power quality monitoring. *IEEE Access*, 10:91963–91975, 2022.

Andy Liaw, Matthew Wiener, et al. Classification and regression by randomforest. *R news*, 2(3):18–22, 2002.

Hui Liu, Fida Hussain, Yue Shen, Sheeraz Arif, Aamir Nazir, and Muhammad Abubakar. Complex power quality disturbances classification via curvelet transform and deep learning. *Electric Power Systems Research*, 163:1–9, 2018.

Misael Lopez-Ramirez, Eduardo Cabal-Yepez, Luis M Ledesma-Carrillo, Homero Miranda-Vidales, Carlos Rodriguez-Donate, and Rocio A Lizarraga-Morales. Fpga-based online pqd detection and classification through dwt, mathematical morphology and svd. *Energies*, 11(4):769, 2018.

Prakash K. Ray, Asit Mohanty, and Tapas Panigrahi. Power quality analysis in solar PV integrated microgrid using independent component analysis and support vector machine. *Optik*, 180:691–698, 2 2019. doi: 10. 1016/j.ijleo.2018.11.041. URL `https://linkinghub.elsevier.com/retrieve/pii/S0030402618317868`.

Irina Rish et al. An empirical study of the naive bayes classifier. In *IJCAI 2001 workshop on empirical methods in artificial intelligence*, volume 3, pp. 41–46. Citeseer, 2001.

Gurpreet Singh, Yash Pal, and Anil Kumar Dahiya. Classification of Power Quality Disturbances using Linear Discriminant Analysis. *Applied soft computing*, 138:110181, 5 2023. doi: 10.1016/j.asoc.2023.110181. URL `https://www.sciencedirect.com/science/article/abs/pii/S1568494623001990?via%3Dihub`.

Utkarsh Singh and Shyam Narain Singh. Application of fractional fourier transform for classification of power quality disturbances. *IET Science, Measurement & Technology*, 11(1):67–76, 2017.

S.T Suganthi, Arangarajan Vinayagam, Veerapandiyan Veerasamy, A. Deepa, Mohamed Abouhawwash, and Mariammal Thirumeni. Detection and classification of multiple power quality disturbances in Microgrid network using probabilistic based intelligent classifier. *Sustainable energy technologies and assessments*, 47: 101470, 10 2021. doi: 10.1016/j.seta.2021.101470. URL `https://linkinghub.elsevier.com/retrieve/ pii/S221313882100480X`.

Arangarajan Vinayagam, Veerapandiyan Veerasamy, Padmavathi Radhakrishnan, Maheswari Sepperumal, and Kalaivani Ramaiyan. An ensemble approach of classification model for detection and classification of power quality disturbances in PV integrated microgrid network. *Applied soft computing*, 106: 107294, 7 2021. doi: 10.1016/j.asoc.2021.107294. URL `https://linkinghub.elsevier.com/retrieve/ pii/S1568494621002179`.

Shouxiang Wang and Haiwen Chen. A novel deep learning method for the classification of power quality disturbances using deep convolutional neural network. *Applied energy*, 235:1126–1140, 2 2019. doi: 10.1016/ j.apenergy.2018.09.160. URL `https://linkinghub.elsevier.com/retrieve/pii/S0306261918314703`.

Petros Xanthopoulos, Panos M. Pardalos, and Theodore B. Trafalis. *Linear Discriminant Analysis*. 11 2012. doi: 10.1007/978-1-4419-9878-1\{_}4. URL `https://doi.org/10.1007/978-1-4419-9878-1_4`.

Haihua Xue, Alian Chen, Deqiang Zhang, and Chenghui Zhang. A novel deep convolution neural network and spectrogram based microgrid power quality disturbances classification method. In *2020 IEEE Applied Power Electronics Conference and Exposition (APEC)*, pp. 2303–2307, 2020. doi: 10.1109/APEC39645. 2020.9124252.

Alper Yılmaz, Ahmet Küçüker, Gökay Bayrak, Davut Ertekin, Miadreza Shafie-Khah, and Josep M Guerrero. An improved automated pqd classification method for distributed generators with hybrid svm-based approach using un-decimated wavelet transform. *International Journal of Electrical Power & Energy Systems*, 136:107763, 2022.

Alper Yılmaz, Ahmet Küçüker, and Gökay Bayrak. Automated classification of power quality disturbances in a SOFC&PV-based distributed generator using a hybrid machine learning method with high noise immunity. *International journal of hydrogen energy*, 47(45):19797–19809, 5 2022. doi: 10.1016/j.ijhydene. 2022.02.033. URL https://linkinghub.elsevier.com/retrieve/pii/S0360319922005572.

Ming Zhang, Kaicheng Li, and Yisheng Hu. A real-time classification method of power quality disturbances. *Electric power systems research*, 81(2):660–666, 2 2011. doi: 10.1016/j.epsr.2010.10.032. URL https://www.sciencedirect.com/science/article/abs/pii/S0378779610002671.

