# OpenReview forum: "Machine Learning Based Power Quality Disturbance Classification Under Varying Fundamental Frequency Conditions for Islanding of a Microgrid"
_TMLR — Rejected by TMLR_

### Review · Reviewer_vsXq · 2024-08-24

**Summary Of Contributions:**

The paper investigates the classification of power quality disturbances (PQDs) in microgrids. The main contribution seems to be the fact that authors are actively changing the fundamental frequencies, allowing for fluctuations in their simulations. The main content is a comparative analysis of four basic algorithms, in which the goal is to classify 29 different PQDs in the signals (claimed to be the standard method in this field of study). The results are somewhat as expected, i.e., that noise negatively impacts classification accuracy.

**Audience:**

No

**Claims And Evidence:**

No

**Requested Changes:**

**Why Weakness 1:** I am not an expert in this particular field, but I believe I understood the paper quite well. Indeed, it seems like an interesting study and important contribution for its own field; but the methodology used is textbook level. From a machine learning perspective, I do not believe the paper is of interest to a substantial group of TMLR readers. Here are some ideas that may help bring this work closer to the TMLR audience (or a combination of them):

- Providing a public benchmark with a clear documentation of the code to create simulation datasets for this problem. I do not believe a simple addition of frequency variation is sufficient here, the authors should include some new insights that require advanced approaches to tackle. As authors noted, even basic algorithms can handle the current simulations (though I am skeptical, see below for why weakness 2).

- Testing out of distribution abilities: Authors may want to take a different approach and test OOD samples. For example, train on one range of frequencies and test on another. Train on one noise level and test on another. Which approaches were truly robust? For example, LDA would likely not be robust in more complex noise conditions, since it assumes Gaussian noise (which seems to be the noise type in simulations, so it makes sense LDA performs well in feature extraction).

**Why weakness 2:** It seems that the authors are doing a random train-test split. This is inappropriate for time-series data. It can easily lead to information leakage and inflated accuracies. One appropriate approach would be to design train and test sets by creating two distinct chunks in time (for example, train: 0-2500, test: 2501-3200, or train: 300-2800, test: 0-300 and 2800-3200). Even better approach, the authors may just want to simply create two different datasets (with distinct trials) to prevent any leakage at all.

Overall, the manuscript as it stands is at most a graduate level textbook problem. I do not believe it meets the bar for publication at TMLR, which does not require novelty. Yet, for publication in a machine learning journal, this manuscript should provide an interesting insight for its audience. In its current form, the insight is that "adding noise decreases accuracy in this particular problem." This is, unfortunately, not an interesting insight.

**Strengths And Weaknesses:**

**Strengths**

- The question of interest seems to be relevant to real life applications, and quite important in its own field.
- The paper is well organized, the ideas are simple, and experiments are performed according to field standards.

**Weaknesses**

- I do not believe the work is in scope for TMLR.
- The methodology seems to have a fatal error.

---

### Review · Reviewer_S8nL · 2024-09-20

**Summary Of Contributions:**

The paper evaluates the performance of several classical classifiers on a synthetic data set that was generated to include a mixture of frequencies. The paper suggests using Linear Discriminant Analysis for feature learning on such data.

**Audience:**

No

**Broader Impact Concerns:**

No concerns.

**Claims And Evidence:**

Yes

**Requested Changes:**

The paper needs to make a scientific contribution to machine learning to be in scope for a machine learning journal.

**Strengths And Weaknesses:**

The main weakness of this paper is that I don't see any takeaways pertaining to machine learning: the performance of several off-the-shelf methods on a new data set does not seem to be telling us much about either the methods themselves or machine learning more broadly. Indeed, the data generation process itself is not even new to this work, only the use of a mixture of range of frequencies. Even for an applications paper, it does not seem to raise or tackle any novel issues. In summary, the paper does not appear to be in scope for TMLR.

---

### Decision · Action_Editor_yHnq · 2024-10-10

**Recommendation:** Reject

**Comment:**

This manuscript considers the application of machine learning to an applied problem -- the classification of "power quality disturbances" in microgrid systems. This is a critical task in the design of power distribution systems that integrate with renewable energy sources. The authors discuss this problem, design a dataset to evaluate machine learning approaches to this task, and compare the performance of several off-the-shelf machine learning methods -- including both classifiers and preprocessing procedures such as dimensionality reduction -- on this dataset.

For publication at TMLR, the findings of submitted material must:

- be supported by accurate, convincing and clear evidence, and
- be of interest to at least a subset of TMLR's audience.

For the first of these criteria (evidence), one reviewer pointed out a "fatal flaw" in the design of the empirical study in the submitted manuscript that appears to invalidate the findings of that experiment. Although the reviewer provided some pointers for how the design could be improved, it would require rerunning the experiment from scratch to address, changes that are well outside the scope of minor revision.

Judging the second of these criteria (interest to TMLR's audience) for this manuscript requires some care. Applications of machine learning to real-world problems are of course at least potentially within scope for publication of TMLR. However, the reviewers found that the scope of the study central to this work was not likely to be above the threshold for interest to TMLR's audience, even if the experimental design problem problems raised by one reviewer were to be addressed. The reviewers agreed that the straightforward application of off-the-shelf machine learning methods to a given problem is not in itself of sufficient interest unless the findings can provide _insight_ for TMLR's audience. That does not appear to be the case here.

Although this work does not appear to be a good fit for TMLR, I do want to commend the authors for their efforts in bringing machine learning tools to bear on this important problem.

**Audience:**

No, there does not appear to be sufficient interest in the findings of this paper among the TMLR audience.

**Claims And Evidence:**

No. One reviewer pointed out a "fatal flaw" (to quote reviewer vsXq) in the design of the central empirical study in this work that appears to invalidate the findings drawn from the experiment. Namely, the authors choose to use a randomized train/test split design that is inappropriate for time series data.